# Quasi-Freeform Metasurfaces for Wide-Angle Beam Deflecting and Splitting

**DOI:** 10.3390/nano13071156

**Published:** 2023-03-24

**Authors:** Qiuyu Zhang, Dingquan Liu, Sheng Zhou, Gang Chen, Junli Su, Leihao Sun, Yunbo Xiong, Xingyu Li

**Affiliations:** 1Shanghai Institute of Technical Physics, Chinese Academy of Sciences, Shanghai 200083, China; 2School of Physical Science and Technology, ShanghaiTech University, Shanghai 200031, China; 3School of Optoelectronics, University of Chinese Academy of Sciences, Beijing 100049, China

**Keywords:** metasurface, wide-angle, beam deflecting, beam splitting, inverse design

## Abstract

Metasurfaces attracted extensive interests due to their outstanding ability to manipulate the wavefront at a subwavelength scale. In this study, we demonstrated quasi-freeform metasurfaces in which the radius, location, and height of the nanocylinder building blocks were set as optimized structure parameters, providing more degrees of freedom compared with traditional gradient metasurfaces. Given a desired wavefront shaping objective, these structure parameters can be collectively optimized utilizing a hybrid optimized algorithm. To demonstrate the versatility and feasibility of our method, we firstly proposed metasurfaces with deflecting efficiencies ranging from 86.2% to 94.8%, where the deflecting angles can vary in the range of 29°–75.6°. With further study, we applied our concept to realize a variety of high-efficiency, wide-angle, equal-power beam splitters. The total splitting efficiencies of all the proposed beam splitters exceeded 89.4%, where a highest efficiency of 97.6%, a maximum splitting angle of 75.6°, and a splitting uniformity of 0.33% were obtained. Considering that various deflecting angles, and various splitting channels with different splitting angles, can be realized by setting the optical response of metasurfaces as the optimization target, we believe that our method will provide an alternative approach for metasurfaces to realize desired wavefront shaping.

## 1. Introduction

Beam deflectors and splitters play a critical role in various optical and photonic systems, such as optical holography, spectroscopy, interferometers, sensing, and optical communications [1,2,3,4]. Traditional optics for beam deflecting and splitting, such as lenses and triangular prisms, are bulky and heavy, which limits their applications in compact optical systems [5,6]. In recent years, metasurfaces emerged as a promising new device to replace or complement their traditional optical elements in the compact optical systems [7,8]. Metasurfaces are two-dimensional planar artificially constructed structure composed of subwavelength metallic or high-refractive-index dielectric antennae. By spatially adjusting the geometrical parameters of the building blocks, the phase, amplitude, and polarization of the incident light can be tuned at a subwavelength resolution [9,10]. Due to their advantages of ultra-thin thickness and versatile functionalities in a planar structure, metasurfaces attracted great interests in recent years. Various optical components, including flat lenses [11], holograms [12], polarization optics [13], beam deflectors [14], and beam splitters [15], were demonstrated.

For a large portion of metasurfaces, the geometrical parameters of the building blocks are determined by forward design method [16,17,18]. A typical workflow of the forward design method includes three steps. First, the period and height of the building blocks are pre-designed based on the physical intuition and the experience of researchers. Then, a meta-library of physically intuitive building blocks, such as nano-rings, nanocylinders, and nanorods, is built through parameter sweeping to cover the required 0 to 2π phase shifts. Finally, a finite number of building blocks with desired phase shifts are collected from the pre-built meta-library and arranged with periodicity to realize targeted wavefront shaping. The forward design approach is widely used due to its interpretable framework and ease of implementation [19,20]. However, as the design constraints and design degrees of freedom scale up, it becomes less useful and may result in limit device performance. For example, metasurface-based beam deflectors and beam splitters are composed of periodic supercells, where the diffraction angle of each diffraction channel are determined by the supercell period. For such a metasurface designed by forward design method, the supercells are composed of periodic building blocks and the period of the supercell can only be adjusted by manipulating the numbers of the building blocks. Obviously, once the targeted splitting angles are changed, the period of the building block should be reset and the meta-library should be rebuilt accordingly, resulting in limited design flexibility. Moreover, metasurfaces designed by the forward design strategy, achieving high efficiency beam deflecting and beam splitting at a small angle, have a dramatic efficiency drop for a wide angle [21]. A large deflecting or splitting angle can provide a wide field of view. Consequently, the low efficiency at wide angle may limit their further applications in optical systems. The efficiency drops are mainly caused by two reasons. The first one is the insufficient design degrees of freedom. Additionally, the second reason is that the near-field interaction between adjacent building blocks is not considered in the forward design, resulting in the inaccurate prediction of the metasurfaces [22]. 

Alternative approaches for realizing high efficiency deflecting and splitting with wide angle are based on inverse design concept. The inverse optimization methods brought promise to metasurfaces beyond the capabilities of traditional methods, such as broadband response, unconventional electronic logic gate, multiplexed response, large numerical apertures, and high efficiencies [23,24]. In the inverse design methodology, the desired optical response was framed as a figure of merit (FOM), and the optimization algorithm was utilized to optimize the structure parameters to maximize or minimize the figure of merit through iterative computations. The inverse designed methods were also applied to explore metasurfaces for wide-angle beam deflecting. For instance, the freeform metasurfaces-based grating [21,25] and catenary-like metasurfaces [26,27] designed by topological optimization method were demonstrated to realize wide-angle high efficiency beam deflecting. However, the catenary-like metasurfaces may not always yield the global optimal solution and a relatively low deflection efficiency of 76% was obtained at a 67.3° angle [27]. A high deflection efficiency of 89% at 75° angle was obtained in freeform metasurface [25]. However, it may take hundreds of iterations to obtain the optimal structure due to the immense degrees of freedom in the freeform space. Hence, a design strategy that can realize high deflecting performance with less-consuming time is needed. As for the metasurface-based beam splitters, recent works were mainly conducted by utilizing forward design method to improve the splitting performance of gradient metasurfaces [15,28,29,30,31,32,33,34]. Among these beam splitters, a splitting efficiency of 93.4% was achieved, while the splitting angle was only 50° [34]. Few works were devoted to obtain high efficiency at wider angle by inverse design method. Therefore, it is necessary to investigate beam splitters with larger splitting angle and efficiency by inverse design. 

In this paper, we proposed quasi-freeform metasurfaces to realize wide-angle, high efficiency beam deflecting and splitting. The supercells inside the metasurfaces were defined as an ensemble of simple nanocylinders that were collectively optimized by a hybrid optimization algorithm. The radius and location of each nanocylinder, and the height of the supercells, were optimized structure parameters. To illustrate the feasibility and versatility of the proposed method, we begin with demonstrating six metasurfaces for deflecting incident light to +1 channel with deflecting angles ranging from 29° to 75.6°. The deflecting efficiencies were higher than 86.2% and the highest deflecting efficiency reached 94.8% at 56.4° after optimization. With further study, we applied our concept to realize a variety of wide-angle, high efficiency, equal-power beam splitters. In the design of the beam splitters, we started by optimizing metasurfaces with three port splitting output with the maximum splitting angles ranging from 29° to 75.6° and splitting efficiencies ranging from 89.4% to 95.7%. Additionally, metasurface with five port splitting output was then proposed, where the overall efficiency and splitting uniformity were 93.6% and 5.5%, respectively. The beam splitters mentioned above adopted the asymmetrical structure to generate more degrees of freedom. Apart from the asymmetrical structure, we also proposed metasurfaces with symmetrical structure to realize beam splitting. By adopting symmetrical structure with a smaller number of structure parameters, we further addressed metasurfaces for generating five and seven port splitting output. In the two optimized metasurfaces with a maximum splitting angle reaching 75.6°, the total splitting efficiencies reached 97.6% and 95%, and the splitting uniformity were 2.7% and 4.7%, respectively. Those cases studies proved that various deflecting angles, and various splitting channels with different splitting angles, can be realized by setting the optical response of the quasi-freeform metasurfaces as the optimization target. In general, the inversely designed quasi-freeform metasurfaces provided useful insights into the design of metasurfaces for realizing desired wavefront shaping, especially for realizing wide-angle deflecting and splitting. 

## 2. Materials and Methods

The quasi-freeform metasurfaces were composed of periodic supercells, as illustrated in Figure 1a, where the black dashed lines depict the boundary of the supercells. The upper and lower panels were corresponding to metasurfaces for beam deflecting and splitting, respectively. The supercells were composed of a high refractive index dielectric nanocylinder array sitting on a SiO_2_ substrate. The refractive index of Si and SiO_2_ were $n_{\mathrm{Si}}=3.47$ and $n_{{\mathrm{SiO}}_{2}}=1.44$, respectively, at an operation wavelength of 1550 nm. The performances of metasurfaces were determined by the optical response of the supercells. The deflecting and splitting angles were calculated according to grating theory [35]:(1)$$P_{X}\left(\sin\theta +\sin\theta_{n}\right)=n\lambda ,$$
where $P_{X}$ is the period length of the supercell in *x* direction. θ and $\theta_{n}$ are the incident angle and the diffraction angle of the *n* channel, respectively. λ is the working wavelength. When light is normally incident to the grating, the diffraction angle of the *n* channel can be calculated by:(2)$$\theta_{n}=\sin^{-1}\frac{n\lambda }{P_{X}},$$

Based on the above equations, different deflecting or splitting angles designs can be obtained by varying the period length $P_{X}$ of the supercell. Meanwhile, the period length in *y* direction was set to 640 nm to allow only zeroth-order diffraction in y direction. 

Supercells with asymmetrical and symmetrical structure were proposed in this paper. The schematics of the supercells are shown in Figure 1b, where the upper and middle panels are the top view of the supercells, and the lower panel illustrates the side view of the supercells. The red dashed lines and black dashed lines depict the center locations of the supercells and the nanocylinders, respectively. The radius ($R_{i},\;i=1,2,3,\ldots$), center positions ($d_{i},\;i=1,2,3,\ldots$), and the height (H) of the nanocylinders are structure parameters that can be optimized for the optimal optical response. When the supercell was composed of M numbers of nanocylinders, the numbers of the optimized structure parameters for the asymmetrical and symmetrical structure were set to 2M + 1 and M + 2, respectively. The number of the optimized structure parameters in the supercell for traditional metasurfaces was set to M, where the location and height of the building blocks were not set as optimized structure parameters. Comparing to those traditional metasurfaces, the metasurfaces proposed here provided more degrees of freedom. The parameter space for such a supercell increased as the number of the optimized structure parameters also increased. Meanwhile, the supercells will generate multi-channel diffraction channels, where the diffraction efficiency of each channel cannot be independently controlled and was simultaneously affected by the structure parameters. The parameter space for such a supercell can be enlarged exponentially, making the tradition forward design a daunting task. Hence, the optimization-based inverse design methodology is introduced to optimize the structure parameters. 

The optimization-based inverse design methodology consisted of an electromagnetic solver and an objective optimizer based on algorithm, as shown in Figure 2. The electromagnetic solver, carried out by utilizing 3D finite-domain time-domain (FDTD) by the commercial software (Lumerical Solutions, Vancouver, BC, Canada), was used to calculate the optical response of the supercell. Hence, the near field coupling effect between adjacent nanocylinders was considered in the optimization, which lead to a more accurate prediction of the metasurface’s optical response. In the simulation, the plane wave was normally incident from the underneath of the substrate. Periodic boundary conditions are applied in both *x* and *y* directions. The perfectly matched layers were used along the incident *z* direction. The efficiency of each channel was obtained from the grating transmission analysis group which was set on the *x*-*y* plane. 

The backward optimizer can be realized by any optimization algorithms, such as genetic algorithm (GA) [20,36], topological optimization (TO) [21], deep neural network (DNN) [37], and particle swarm optimization (PSO) [38,39], etc. PSO is a heuristic and stochastic technique, which originated from imitating the schooling behavior of birds when searching for food [40]. Due to the deficiency of diversity, this algorithm is easily trapped in premature convergence, especially in global searching for the problems with high nonlinearity. Incorporating PSO with other algorithms can help to prevent the premature convergence. Different hybrid algorithms were adopted to solve many specific engineering optimizing problems. Here, we incorporated PSO with genetic algorithm (HGA-PSO) [41] for the inverse design algorithm-assisted parameter space searching. In the optimization, the lower and upper values of the nanocylinder radius were 50 nm and 270 nm. Additionally, the height of the nanocylinders can be optimized in the range of 530 nm to 830 nm. The location variation range of each nanocylinder was determined by the periodic lengths of the supercells and the nanocylinders number. For example, if there were two nanocylinders within a supercell with a supercell length of $P_{X}$ in *x* direction, the locations of the nanocylinders were set to meet $-P_{X}/2<d_{1}<0$ and $0<d_{2}<-P_{X}/2$, where $d_{1}$ and $d_{2}$ were the center location of the two nanocylinders along x direction. Meanwhile, the algorithm parameters, such as populations number, crossover rate, variation rate, weight coefficients of the PSO, etc., were also initialized in the algorithm. 

## 3. Results and Discussion

In this paper, the structure parameters of the metasurfaces were optimized for realizing two kinds of optical responses. Thus, two equations were required to define the figure of merit (*FOM*) functions. In the design of beam deflectors, the goal was to deflect the incident light to a selected diffraction channel with high efficiency. To realize high beam deflecting efficiency, we defined the figure of merit as following equation: (3)$$FOM_{d}=1-\eta_{n},$$
where $\eta_{n}$ is the deflecting efficiency of the selected channel *n*. As for the design of beam splitters, the ultimate target was to equally split the incident light into selected diffraction channels with high efficiency. Therefore, to optimize the overall splitting efficiency and the splitting uniformity simultaneously, we defined the figure of merit as:(4a)$$FOM_{s}=\alpha *\left(1-\eta_{total}\right)+\beta *\sigma ,$$
(4b)$$\eta_{total}=\sum \eta_{n},$$
(4c)$$\sigma =\frac{\eta_{max}-\eta_{min}}{\eta_{max}+\eta_{min}}.$$
where $\eta_{total}$ represents the overall splitting efficiency of the selected splitting channels. σ points to the splitting uniformity among the desired channels. $\eta_{max}$ and $\eta_{min}$ are the maximum and minimum efficiency among selected channels. Additionally, α and β are the weight coefficients. 

### 3.1. Metasurfaces for Beam Deflecting

Firstly, we addressed a set of metasurfaces that can deflect the normally incident light to +1 channel with deflecting angles ranging from 28.97° to 75.64°. Six supercells, named D1, D2, D3, D4, D5, D6, were designed by the inverse design methodology in Figure 2. The deflecting angle can be tuned by adjusting $P_{X}$, as illustrated in Equation (2). $P_{X}$ were set to 3200 nm, 2620 nm, 2160 nm, 1860 nm, 1700 nm, and 1600 nm, respectively. The near field coupling effect between adjacent nanocylinders may reduce the transmission efficiency and result in low deflection efficiency. To minimize the near field coupling effect, sufficient spacing was required between adjacent nanocylinders. Therefore, the number of the nanocylinders should be set properly, especially when $P_{X}$ is relatively small. Here, the numbers of nanocylinders M were set to 4, 3, 3, 2, 2, and 2 for the six supercells. The nanocylinders were arranged to form the asymmetrical supercell, generating 2M + 1 structure parameters for optimization in each supercell. The figure of merit in Equation (3) was adopted for optimization, in which $\eta_{n}$ represents the deflecting efficiency of +1 channel. Optimization trajectories of the supercells are depicted in Figure 3a. The figure of merit converged with different speed in the six cases. The convergence speed was related to the number of optimized structure parameters. In general, the optimizations find and yield the optimal results within 15 iterations in all cases. Figure 3b summarizes the optimized deflecting performances of the optimized metasurfaces. The deflecting efficiencies were larger than 90% for deflection angles ranging from 28.97° to 65.75°. Meanwhile, the highest deflecting efficiency of 94.8% was obtained for 56.44° deflection. Moreover, the deflection efficiency maintained 86.2% for 75.64° deflection. Therefore, we successfully demonstrated quasi-freeform metasurfaces that can deflect light into different angles with high efficiencies. 

Figure 3c depicts the phase distribution profiles of electric field Ex in the x-z plane for the optimized metasurfaces under x-polarized light. The black arrows show the propagation directions of the incident and deflected light. The white lines mark the boundaries of the nanocylinders and the substrate. Due to the different values of $P_{X}$, the numbers of periodic supercells in the simulation regions were set differently to intuitively display the different beam deflecting angles. The black dashed lines show the side view of the optimized supercell. As observed in Figure 3c, the wavefronts of the incident light were nearly perpendicular to the x axis before passing through the metasurfaces. After being modulated by the metasurfaces, the wavefronts of the incident light were clearly tilted at different angles with respect to the y axis. The wavefronts of the deflecting light were not an ideal parallel incline plane due to the disturbance of the unselected channels. The total efficiencies of the unwanted channels $\eta_{u}$ were 2.44%, 1.48%, 1.42%, 2.59%, 3.58%, and 0.56% for the optimized metasurfaces. Meanwhile, the remnant optical powers went into reflection, as observed in Figure 3c. By setting the frequency-domain field profile monitors underneath the incident light, a reflectance of 3.69%, 4.12%, 5.29%, 2.56%, 3.94%, and 13.57% were obtained. 

To demonstrate the advantages of the proposed beam deflectors based on our method, a comparison between some previously reported studies and the proposed beam deflectors are listed in Table 1. The beam deflectors proposed by Yu et al. [14], Egorov et al. [42], and Aoni et al. [43] were based on forward design gradient metasurfaces. Among these gradient metasurfaces, the highest efficiency of 95% was obtained. However, the deflecting angle was only 8.7°. Topology-optimized catenary-like metasurfaces [26,27] and freeform metasurfaces [21,25] were demonstrated to achieve a higher deflecting performance. Compared to the gradient metasurfaces and topology-optimized catenary-like metasurfaces, our structures showed a superior performance in deflecting efficiency and deflecting angle. The deflecting efficiency of freeform metasurfaces (89%) was slightly higher than that of quasi-freeform metasurfaces (86.2%) at an angle of around 75°. However, hundreds of iterations are required for freeform metasurfaces to obtain the final structure due to the immense degrees of freedom in the freeform space. Due to the moderate degrees of freedom in quasi-freeform metasurfaces, we can find and yield the optimal structures within 15 iterations, which is less time-consuming. Therefore, the inversely designed quasi-freeform metasurfaces may provide an efficient way to realize wide-angle beam deflecting.

### 3.2. Metasurface for Beam Splitting

Apart from the optimization of metasurfaces for beam deflecting, our optimization strategy can also readily generalize to the design of high efficiency, equal power beam splitting. In order to further verify the feasibility and versatility of the proposed quasi-freeform metasurfaces, we demonstrated several sets of metasurfaces for three, five, and seven port splitting output. In the design of beam splitting, two types of supercells including asymmetrical and symmetrical structure were utilized. Both structures obtained high splitting performance after optimization. 

#### 3.2.1. Asymmetrical Quasi-Freeform Metasurfaces for Beam Splitting

To reveal that different wavefront shaping tasks can be realized by modulating the figure of merit function, six supercells mentioned in Section 3.1 were optimized to realize three port splitting output. For clarity, the supercells were named as S1, S2, S3, S4, S5, and S6. The period lengths $P_{X}$ and the nanocylinders numbers of the six supercells were the same as supercell D1, D2, D3, D4, D5, and D6 in Section 3.1. To optimize such gratings with equal power splitting, we adopted the figure of merit in Equations (4a)–(4c), where the weight coefficients α and β were set to 0.6 and 0.4, respectively. Optimization trajectories of the supercells are displayed in Figure 4a. The optimizations yielded six optimal structures after 20 generations. The splitting efficiency of the selected channels and the splitting uniformity σ of the six optimized structures are depicted in Figure 4b. The splitting efficiencies of the supercells ranged from 89.4% to 95.7%. The largest efficiency reached 95.7% in supercell S3, where the splitting angle was 45.86°. A splitting angle of 75.64° was obtained in supercell S6, with a splitting efficiency reaching 89.4%. Meanwhile, the splitting uniformity σ of the optimized structures were 0.78%, 0.33%, 1.57%, 0.94%, 0.96%, and 0.84%, respectively, which illustrate good uniformity of the splitting channels. To illustrate the performances of the metasurfaces, we calculated the detailed efficiency at each diffraction channel and presented them in Figure 4c. The efficiencies only showed slight differences among selected channels, indicating a lower residual variance in the optimized structures. It is worth noting that there still exist unselected higher diffraction channels in supercell D1. The diffraction efficiencies of the unselected −2 and +2 channels in supercell D1 were 1.2% and 1.4%, respectively. Meanwhile, a reflectance of 7.6%, 4.8%, 4.3%, 7.7%, 5.2%, and 10.6% were obtained in the optimized structures. Therefore, we successfully demonstrated quasi-freeform metasurfaces that can split light into three channels with different angles. The only difference between the optimizations of these six metasurfaces for three port splitting output and the optimizations of six beam deflecting in Section 3.1 was the utilization of different figure of merit functions. Consequently, we showed that different wavefront shaping tasks can be realized by framing the optical response as a figure of merit, which proved the versatility of the method proposed in this paper.

To further indicate the feasibility of our method, we addressed metasurface for generating wide-angle five-port splitting output. Here, the undesired ±2 channels in supercell S1 were set as the selected channels, generating five splitting channels with maximum splitting angle reaching 75.64°. The supercell, named supercell S7, also contained four nanocylinders. The optimization structure parameters number was set to nine, which was the same as supercell D1 and S1. We framed the optical response of the supercell as the figure of merit in Equations (4a)–(4c), where the $\eta_{total}$ was the overall efficiency of the five selected channels. Optimization trajectory of the supercell was depicted in Figure 5a. The optimization algorithm yielded the optimum structure after 30 iterations. The illustration in Figure 5a was the schematic of the optimized supercell. The phase distribution profiles of electric field Ex in the x-z plane for the optimized metasurfaces under x-polarized light are shown in Figure 5b. The white lines depict the boundaries of the nanocylinders and substrate. The wavefront of the transmitted light was very different from the wavefront of supercell D1 in Section 3.1. The near field phase distribution in Figure 5b will generate five splitting channels with nearly equal power in the far field. The far field distribution of the splitting channels is depicted in Figure 5c, where the yellow circles with different radius indicate the different angles. The splitting efficiency at each channel was also calculated and depicted in Figure 5d. Compared with supercell S1 in Figure 4c, where the maximum splitting angle was 28.97°, a maximum splitting angle of 75.64° was achieved in supercell S7. Meanwhile, a total splitting efficiency $\eta_{total}$ of 93.6% was achieved, which was slightly higher than the value of supercell S1 (89.4%). The maximum and minimum efficiency were 20.2% and 18.05%, respectively, generating a uniformity error σ of 5.5%. We set a frequency-domain field profile monitor underneath the incident light to obtain the reflectance, and a reflectance of 6.4% was obtained in supercell S7. In general, we successfully demonstrated quasi-freeform metasurfaces with symmetrical structure for generating high efficiency five-port equal power beam splitting with the maximum splitting angle reaching 75.64°.

#### 3.2.2. Symmetrical Quasi-Freeform Metasurfaces for Beam Splitting

In addition to the asymmetrical structure, we, furthermore, proposed quasi-freeform metasurfaces that adopted the symmetry structure, which can also realize high efficiency and equal power beam splitting while using less optimized parameters. As shown in Figure 1b, in a symmetrical supercell, a nanocylinder was set on the center location of the supercell, and the other four nanocylinders were placed symmetrically on both sides of the center nanocylinder. By adopting the symmetrical structure, the optimized structure parameters can decrease from 2M + 1 to M + 2 for a supercell that contained M number of nanocylinders. As a case study, two metasurfaces for high performance beam splitting were demonstrated. 

The first symmetrical supercell, named supercell S8, contained five nanocylinders. The period length $P_{X}$ of supercell S8 were set to 3200 nm, which was the same as supercell D1, S1, and S7 mentioned above. Due to the symmetry, the number of the optimized structure parameters was set to 7. The boundary conditions in x direction were set to be antisymmetric under x polarized light, which will reduce the simulation time. We also adopted the figure of merit in Equations (4a)–(4c). The optimization trajectory of metasurfaces is depicted in Figure 6a. The optimization yielded the optimum supercell in 25 iterations. The illustration in Figure 6a is the schematic of the optimized structure. The phase distribution profiles of electric field Ex in the x-z plane for the optimized metasurfaces under x-polarized light are depicted in Figure 6b. The white lines depict the boundaries of the nanocylinders and substrate. The phase distribution of the electric field of Ex are symmetrical due to the symmetry of the structure. The far field distribution of the splitting channels for the supercell is depicted in Figure 6c. The efficiency at each splitting channel was calculated and depicted in Figure 6d. The splitting efficiencies distributions were clearly symmetrical with respect to the zero-order due to the symmetry of the structure. For supercell S8, a total splitting efficiency $\eta_{total}$ reaching 97.6% and splitting uniformity σ as low as 2.7% was obtained in the optimization metasurface. In general, we successfully demonstrated quasi-freeform metasurfaces with symmetrical structure for five port splitting output with the maximum splitting angle reaching 75.64°.

The second symmetrical supercell had a supercell period $P_{X}$ of 4800 nm, which will generate seven splitting channels with maximum angle reaching 75.64°. The supercell, named supercell S9, also contained five nanocylinders. The number of the optimized structure parameters was set to seven. We also defined the figure of merit as Equation (4a)–(4c) and the figure of merit during the optimization is depicted in Figure 7a. The optimization can quickly yield an optimum result in 25 iterations. The illustration in Figure 7a depicts the side view of the optimized supercell S9. We also simulated the phase distribution profiles of electric field Ex in the x-z plane and present it in Figure 7b. The phase distribution of the electric field of Ex was also symmetrical with respect to the y axis. The far field distribution of the seven splitting channels was depicted in Figure 7c. The splitting angles of the seven channels were consistent with the calculated values in Equation (2). Figure 7d shows the splitting efficiency at each splitting channel. The splitting efficiencies distributions were also symmetrical with respect to the zero-order. A total splitting efficiency $\eta_{total}$ was 95%. The maximum and minimum splitting efficiencies were 14.24% and 12.97%, respectively, generating a splitting uniformity σ of 4.7% in the output beam arrays. Therefore, quasi-freeform metasurfaces with symmetrical structure for seven-port splitting output with the maximum diffraction angle reaching 75.64° was realized.

#### 3.2.3. Comparison of Our Works to Previously Reported Beam Splitters 

To reveal the diffraction performances of the inversely designed quasi-freeform metasurfaces, a comparison between the previously reported metasurface-based beam splitters and the proposed beam splitters is presented in Table 2. Specifically, we compared the splitting efficiency and splitting angle in each work. Previously reported beam splitters were forwardly designed based on gradient metasurfaces [15,28,29,30,31,32,33,34], which can be implemented easily, although at the cost of low efficiency and small angle. Among these works, a highest splitting efficiency and maximum splitting angle were achieved by Liu et al. [34], where the splitting efficiency and angle were 93.4% and 50°, respectively. Meanwhile, previous researchers mainly made efforts in achieving variable split ratio or polarization control. We focused on the enhancement of the efficiency for wide-angle beam splitting. In the proposed beam splitters based on inversely designed quasi-freeform metasurfaces, the splitting angle can be flexibly tuned in the range from 29° to 75.6°. Moreover, a maximum splitting angle of 75.6° and a splitting efficiency of 97.6% was obtained for five-port splitting output. In general, the proposed quasi-freeform metasurface-based beam splitters displayed substantially improved efficiency and splitting angle compared to the previously reported beam splitters.

Some previous reported metasurfaces can realize polarization-insensitive beam splitting. The beam splitters in this paper could not always maintain high splitting performance under y-polarization incident. Appendix A illustrates the splitting efficiency and splitting uniformity of optimized supercells for three-port splitting output. The supercells in this paper contained one array of nanocylinder building blocks. Future work can be carried out by inversely designed supercells that contain two arrays of nanocylinders to achieve polarization insensitive beam splitting.

### 3.3. Broadband and Fabrication Feasibility Analysis

Here, we explored the performance of the proposed beam deflectors and splitters in waveband. We calculated the deflecting and splitting performance of the optimized supercells with a wavelength scanning from 1500 nm to 1600 nm, as illustrated in Figure 8a–c. Clearly, supercell D1, D2, D3, D4, and D5 can maintain high deflecting efficiency in such a wide bandwidth, where the lowest efficiency was higher than 91% with a wavelength between 1530nm to 1570 nm. The efficiency of supercell D6 was higher than 78% in wavelength range of 1530 nm to 1570 nm. The efficiency decreased quickly when the wavelength was greater than 1580 nm. For the wavelength between 1540 nm to 1560 nm, the lowest total splitting efficiencies were 88.9%, 95%, 95.3%, 91.7%, 94.5%, 88.2%, 85.9%, 93.7%, and 94.8%, and the splitting uniformity was lower than 10%, 9.3%, 6.7%, 7.9%, 13.5%, 9.6%, 22.7%, 5.8%, and 23.2%, for optimized supercell S1, S2, S3, S4, S5, S6, S7, S8, and S9, respectively. Overall, the optimized supercells for beam deflectors and splitters had a certain broadband characteristic. 

The fabrication error was unavoidable in practice fabrication. The optimized supercell D1 and D2 were selected as examples for evaluating the effects of radius and height of the nanocylinders on the deflecting efficiency. The results, as depicted in Figure 8d, showed that the supercells can maintain high deflecting efficiency when the height shifts by ±50 nm. Compared with error in height, variation in radius can cause a relatively large efficiency decrease, as depicted in Figure 8e. The deflecting efficiency was larger than 90.5% and 90.7% when the radius shifted by ±5 nm. When the radius shifted by ±15 nm, the efficiencies were only 70.3% and 63.8%, respectively. Consequently, it is pivotal to precisely control the radius of the supercells during manufacturing. The structure parameters of the optimized supercells are listed in Appendix A. Electron beam lithography was reported to realize a narrow resolution below 10 nm [44,45], which can be utilized to fabricate the proposed quasi-freeform metasurfaces. The aspect ratio, defined as the ratios of a structure’s height and minimum feature size, is also a key metric that should be considered in manufacturing process [46]. Figure 8f depicts the height and the aspect ratio of the optimized supercells. Supercell S8 had the highest aspect ratio of 12.2, where the nanocylinder’s height and minimum feature size were 552 nm and 45 nm, respectively. Metasurface with such a high aspect ratio can be fabricated by employing atomic layer deposition [47]. For the rest of the optimized supercells, the aspect ratio was a relatively low value. For instance, the aspect ratios of supercell S5 and S6 were 2.5 and 2.4, respectively. Meanwhile, future work can be implemented by incorporating the aspect ratio into the optimization process, to enhance the robustness criteria of the quasi-freeform metasurfaces in manufacturing.

## 4. Conclusions

In summary, we presented wide-angle, high efficiency beam deflectors and splitters based on quasi-freeform metasurfaces. In the quasi-freeform metasurfaces, the radius, location, and height of the nanocylinder building block in a supercell were set as the optimized structure parameters, which will provide more design degrees of freedom. The quasi-freeform metasurfaces proposed in this paper can combine the features of traditional and freeform metasurfaces. By setting the supercell towards a pre-set angle as the optimization target, various deflecting and splitting angles gratings were inversely designed by a hybrid optimization algorithm. In the design of beam deflectors, the deflection efficiencies ranging from 86.2% to 94.8% with the deflection angles varying in the range of 29°–75.6° were demonstrated. With further study, we demonstrated two kinds of metasurfaces, including asymmetrical and symmetrical metasurfaces, for wide-angle, high efficiency, equal-power beam splitting. In the design of beam splitters based on asymmetrical structure, the splitting efficiencies ranging from 89.4% to 95.7% with the splitting angles varying in the range of 29°–75.6° were obtained. Meanwhile, the splitting uniformities were lower than 5.5%. In the design of beam splitters based on symmetrical structure, a total splitting efficiency reaching 97.6% and 95%, with a splitting uniformity of 2.7% and 4.7%, were achieved for five-port and seven-port splitting output, respectively, where the maximum splitting angles of 75.6° were achieved in both designs. Additionally, broadband analysis indicated that the proposed metasurfaces have a certain broadband characteristic. Compared to traditional metasurfaces, the quasi-freeform metasurfaces provide more degrees of freedom, thus resulting in higher performance. Compared to fully freeform metasurfaces, the structures can avoid immense parameter optimization space while obtaining performances comparable to freeform metasurfaces. Overall, we believe that this work will provide an alternative approach for metasurface to realize desired wavefront shaping.

## Figures and Tables

**Figure 1 nanomaterials-13-01156-f001:**
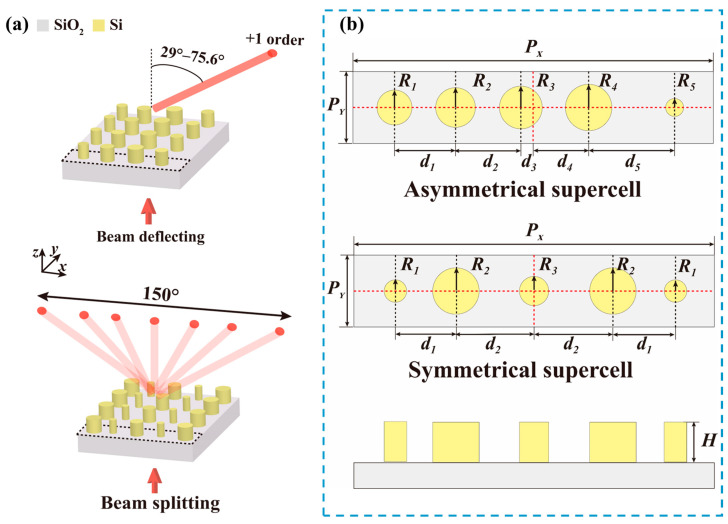
(**a**) Schematics of quasi-freeform metasurfaces for wide-angle beam deflecting (upper panel) and splitting (lower panel); (**b**) schematics of the supercells in quasi-freeform metasurfaces. The upper and middle panels corresponded to the top views of the asymmetrical and symmetrical supercell, and the lower panel illustrates the side view of the supercell.

**Figure 2 nanomaterials-13-01156-f002:**
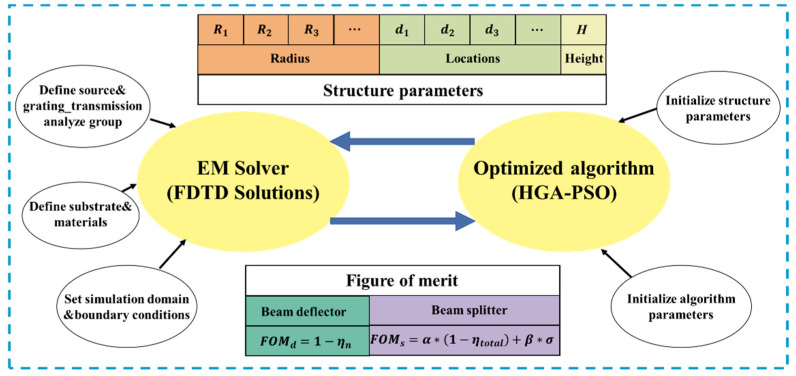
The framework for the optimization-based inverse design methodology.

**Figure 3 nanomaterials-13-01156-f003:**
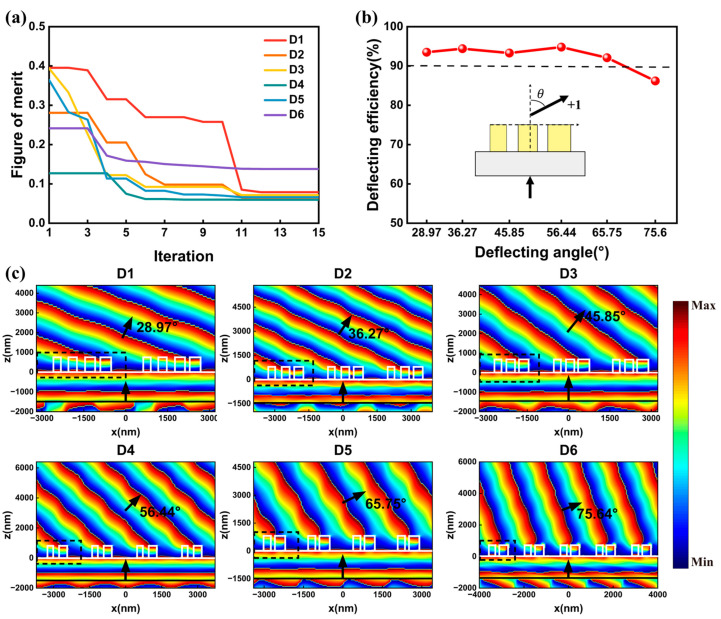
(**a**) The optimization trajectories of metasurfaces for beam deflecting with different angles. (**b**) The deflecting performances of the optimized metasurfaces. The red dotted line depicts the deflecting efficiency for +1 channel with different deflecting angles. (**c**) Phase distribution profiles of Ex in the x-z plane of the optimized metasurfaces under 1550 nm x-polarized incident light. The black arrows represent the propagation directions of the incident and deflected light. The white lines mark the boundaries of the nanocylinders and substrate. The black dashed lines show the side view of the supercell.

**Figure 4 nanomaterials-13-01156-f004:**
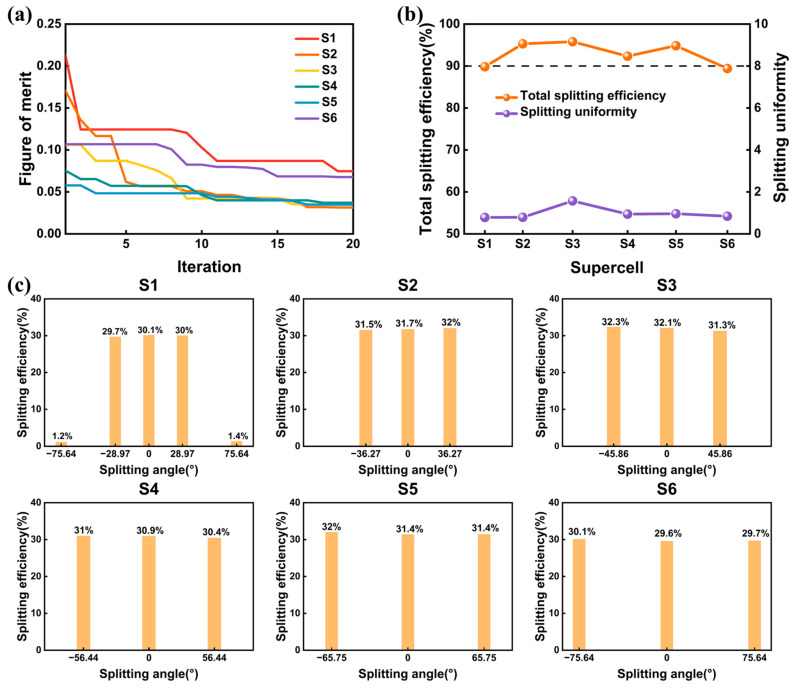
(**a**) The optimization trajectories of metasurfaces for three port splitting output. (**b**) The total splitting efficiency and splitting uniformity σ of the six optimized structures. (**c**) Splitting efficiency at each channel of the optimized supercells.

**Figure 5 nanomaterials-13-01156-f005:**
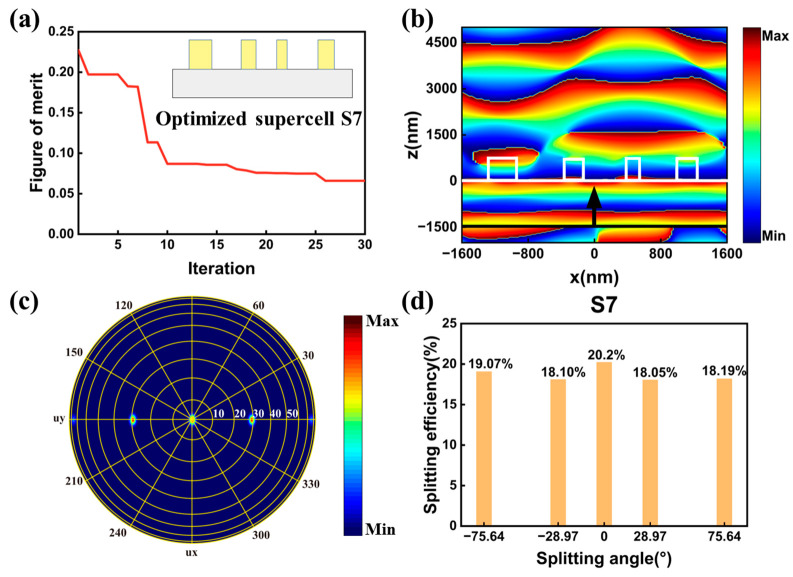
(**a**) The optimization trajectory of metasurface for five-port splitting output. The insert in (**a**) is the schematic of the optimized supercell. (**b**) The phase distribution profile of electric field Ex in the x-z plane for the optimized supercell S7. The white lines depict the boundaries of the nanocylinders and substrate, and the black line and arrow represent the location and direction of the incident light. (**c**) The far field distribution of the five splitting channels. (**d**) The calculated efficiencies at different splitting channels in supercell S7.

**Figure 6 nanomaterials-13-01156-f006:**
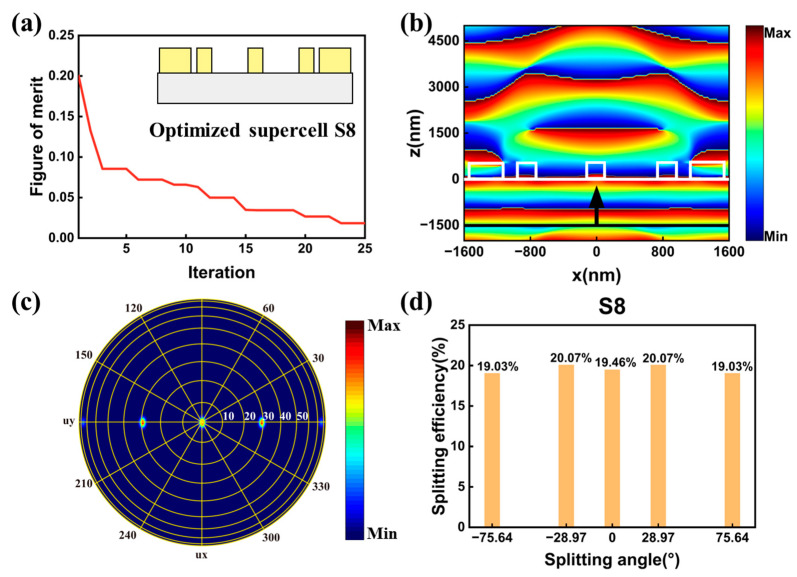
(**a**) The optimization trajectory of metasurface for five-port splitting output. The insert is the side view of the optimized supercell. (**b**) The phase distribution profile of electric field Ex in the x-z plane for the optimized supercell S8. The white lines depict the boundaries of the nanocylinders and substrate, and the black line and arrow represent the location and direction of the incident light. (**c**) The far field distribution of the splitting channels. (**d**) The calculated efficiencies at different splitting channels in supercell S8.

**Figure 7 nanomaterials-13-01156-f007:**
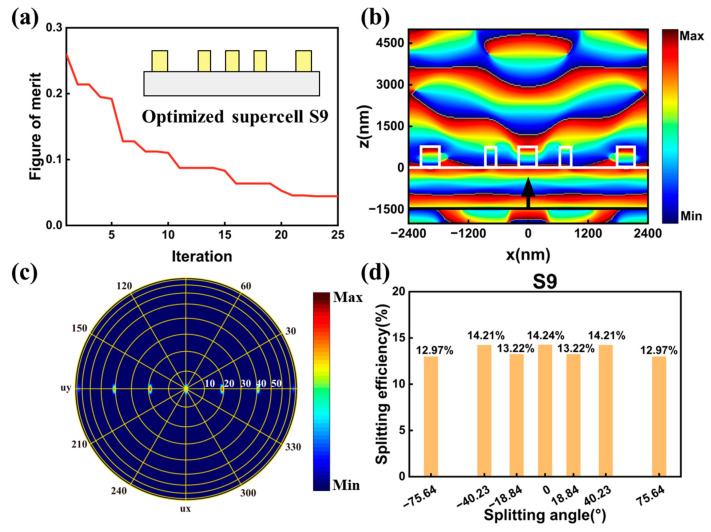
(**a**) The optimization trajectory of metasurface for seven-port splitting output. The insert is the side view of the optimized supercell. (**b**) The phase distribution profile of electric field Ex in the x-z plane for the optimized supercell S9. The white lines depict the boundaries of the nanocylinders and substrate, and the black line and arrow represent the location and direction of the incident light. (**c**) The far field distribution of the splitting channels. (**d**) The calculated efficiencies at different splitting channels in supercell S9.

**Figure 8 nanomaterials-13-01156-f008:**
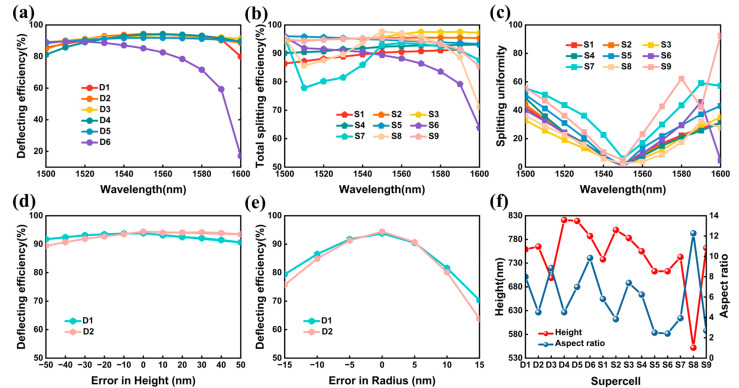
(**a**) Deflecting efficiency of optimized supercells versus the wavelength scanning from 1500 nm to 1600 nm. Total splitting efficiency (**b**) and splitting uniformity (**c**) of optimized supercells versus wavelength in the range of 1500 nm to 1600 nm. The deflecting efficiency of supercell D1 and D2 as a function of the deviation in height (**d**) and radius (**e**) from the ideal design. (**f**) The height and aspect ratio of the optimized supercells.

**Table 1 nanomaterials-13-01156-t001:** Performance comparison of our work and some recent reported works.

Beam Deflector	Wavelength (nm)	Deflecting Efficiency	Deflecting Angle	Description
[14]	705	45%	10°	Gradient metasurfaces
[42]	1550	54%	42.6°	
[43]	715	95%	8.7°	
[26]	1550	65%	67°	Catenary-like metasurfaces
[27]	1550	76%	67.3°	
[21]	1050	84%	75°	Freeform metasurfaces
[25]	1050	89%–95%	10°–75°	
This work	1550	86.2%–94.8%	29°–75.6°	Quasi-freeform metasurfaces

**Table 2 nanomaterials-13-01156-t002:** Performance comparison of our work to previously reported beam splitters based on metasurfaces.

Beam Splitter	Wavelength (nm)	Splitting Efficiency	Splitting Angle
[15]	800	50%	12.2°
[28]	915	71%	49.7°
[29]	996	82%	47.1°
[30]	530	87.2%	14.7°
[31]	532	90%	29.1°
[32]	532	90%	46.8°
[33]	1550	93.2%	17.1°
[34]	690	93.4%	50°
This work	1550	>89.4%	29°–75.6°
		97.6%	75.6°
		95%	75.6°

## Data Availability

Data will be made available on request.

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
