# Peer review of "Quasi-Freeform Metasurfaces for Wide-Angle Beam Deflecting and Splitting"

_nanomaterials, 2023, doi:10.3390/nano13071156_

Round 1

Reviewer 1 Report

All works related to metamaterials and composite materials with an artificially created periodic structure are currently of great relevance. In this paper, the authors propose high-performance wide-angle beam splitters based on quasi-free-form metasurfaces. When designing gratings with different deflection and separation angles, in order to provide more degrees of freedom, the authors used a hybrid optimization algorithm. Deflection efficiency from 86.2% to 94.8% has been demonstrated when deflection angles vary in the range of 29° - 75.6°, which is a very good indicator. The proposed design of the beam splitters provides an overall separation efficiency of 97.6% and 95% with a separation uniformity of 2.7% and 4.7% for the five- and seven-port separation outputs, respectively. This approach results in higher performance and is an excellent alternative to achieve the desired wavefront shaping in a quasi-free-form metasurface. In general, it can be noted that the work is relevant, has scientific and practical value, has novelty and is fully consistent with the theme of the special issue. The work can be accepted for publication after minor modifications, namely: - check the list of references, not all the necessary output data are available everywhere; - the text part on pages 7 and 8 before table 1 almost completely duplicates it in the description, i.e. there is a repetition, maybe somehow transform them; - similarly for table 2 and the text before it in paragraph 3.2.3 ... and also, in my opinion, it is not good that the section ends with a table before the conclusion.

Author Response

 Dear Reviewer,

Thank you for your recognition of our work. We have studied yours’s comments very carefully. Those comments are valuable and very helpful for revising and improving our paper. We have revised our manuscript according to the comments. Changes to our manuscript are all highlighted within the document by using blue colored text. The list of the changes in the manuscript and our point-by-point response to yours’s comments are shown in the cover letter.

We expect that the responses and the corresponding revision of the manuscript will fulfill the requirements of editor and referees for considering this manuscript for publication in “Nanomaterials”. Thank you again for your valuable comments and suggestions. We look forward from you at your earliest convenience.

Reviewer 2 Report

The paper focuses on the application of inverse design methodology in designing metasurfaces for high efficiency beam deflecting and splitting with wide angles.

The paper is in my opinion well-written. It provides a detailed explanation of the forward and inverse design approaches for metasurfaces and highlights their advantages and limitations. The proposed method is also well presented and the specific choices are well substanciated.

Before publication, I would just suggest some minor improvements to the paper.

Firstly, the authors could provide more information on the practical feasibility of their proposed structure, which would help to highlight the potential of their design approach. A brief introduction that cites the most commonly used experimental techniques for fabricating metasurfaces would be useful. To support this, the authors may consider referencing recent reviews on the topic, such as :

Nanomaterials 2021, 11(8), 2079; https://doi.org/10.3390/nano11082079

Nanomaterials 202212(12), 1973; https://doi.org/10.3390/nano12121973

- Particle swarm optimization has been applied to realize nanowire metasurfaces optimized for enhancing light trapping and a similar parameter space has been studied. The authors could consider to cite this paper, if they think it is useful to give strength to their methodology: Luca Zagaglia et al 2022 Nano Ex. 3 021001

- The periodic supercells in fig 1a are  identical. Maybe the optimized cells for the two application should be represented.

Author Response

Dear Reviewer,

Thank you for your recognition of our work. We have studied yours’s comments very carefully. Those comments are valuable and very helpful for revising and improving our paper. We have revised our manuscript according to the comments. Changes to our manuscript are all highlighted within the document by using blue colored text. The list of the changes in the manuscript and our point-by-point response to yours’s comments are shown in the cover letter.The two review papers you mentioned have been referenced in this paper and are marked as [44] on page 13 line 450 and [46] on page 14 line 453. And the number of the third reference  are rectified as [39] on page 5 line 174. We have added a new referece after uploading the cover letter, and  the cover letter can't be  uploaded now. Therefore, we have to modify the correct number here.

We expect that the responses and the corresponding revision of the manuscript will fulfill the requirements of editor and referees for considering this manuscript for publication in “Nanomaterials”. Thank you again for your valuable comments and suggestions. We look forward from you at your earliest convenience.

Reviewer 3 Report

The authors have reported optimization algorithm, used to produce nanomaterial based beam detectors and splitters. Owing to potential great impact of the study on developing materials research and optical technology, the paper would be of interest in the readers of Nanomaterials.

Author Response

Dear Reviewer,

Thank you for your recognition of our work. According to the rest reviewers’ suggestions, we have made some minor revisions on this manuscript. We look forward to hearing the following guidance from you at your earliest convenience.

Reviewer 4 Report

This paper describes quasi-freeform metasurfaces based on silicon and silica. The authors demonstrated wide-angle beam deflecting up to 75.6° with high efficiency > 86.2% by simulation. They also demonstrated light splitting into 5 and 7 beams with high total efficiencies of 97.6% and 95% as well as good uniformity 2.7 % and 4.7%, respectively. It would be of interest to Nanomaterials readers, however, I have some major concerns as follows:

1) The authors should disclose the optimized values of structure parameters Ri, di, and H.

2) It would be informative if the authors mention the polarization dependence. Does this metasurface work for y-polarization?

3) It would be informative if the authors mention the wavelength dependence.

4) It would be informative if the authors mention the sensitivity to fabrication errors. How the deflecting and splitting performances are affected if the size of the structures is slightly varied.

Followings are minor deficiencies:

1) Page 3, Section 2, 1st paragraph. line 5. “Si” and “SiO2” should not be italic but upright.

2) It shall be misleading that the incident direction in Fig. 1 is opposed to that in Fig. 3 and Figs. 5-7.

3) Eq. (4c) shall be incorrect since its numerator and denominator are identical.

4) Page 6, Section 3.1, 2nd paragraph, line 1 (also caption of Fig. 3). “x0z plane” should read “x-z plane".

5) Fig. 4. The upper limit of vertical axis in S6 should be matched with others.

6) “Low uniformity” shall be misleading. It should be reworded to “good uniformity” or “low ununiformity”.

Author Response

Dear Reviewer,

Thank you for your recognition of our work. We have studied yours’s comments very carefully. Those comments are valuable and very helpful for revising and improving our paper. We have revised our manuscript according to the comments. Changes to our manuscript are all highlighted within the document by using blue colored text. The list of the changes in the manuscript and our point-by-point response to yours’ s comments are shown in the cover letter.

We expect that the responses and the corresponding revision of the manuscript will fulfill the requirements of editor and referees for considering this manuscript for publication in “Nanomaterials”. Thank you again for your valuable comments and suggestions. We look forward from you at your earliest convenience.

Round 2

Reviewer 4 Report

I found all the revisions are satisfactory except following minor deficiencies:

1) Section 2, line 5. "of  Si" should be "of Si". (Spaces seem to be redundant.)

2) Page 4, line 7. "issimultaneously" should be "is simultaneously" (A space is needed.)

3) Page 7, line 7 below Figure 3. "performance. ." should be "performance." (Periods are redundant.)

4) Page 12, last line. "Liu et al" should be "Liu et al."

5) Page 13, line 3 from the bottom. "±50 nm" should be "±50 nm". (Not italic but roman.)

Author Response

Dear Reviewer,

Thank you for your carefulness and conscientiousness. We have studied the yours’s comments very carefully and have revised our manuscript according to the suggestions. Changes to our manuscript are all highlighted within the document by using blue colored text. 

Thank you again for your valuable comments and suggestions. We look forward to hearing from you at your earliest convenience.
